# The Stress Concentration Mechanism of Pores Affecting the Tensile Properties in Vacuum Die Casting Metals

**DOI:** 10.3390/ma13133019

**Published:** 2020-07-06

**Authors:** Hanxue Cao, Ziwei Luo, Chengcheng Wang, Jing Wang, Tao Hu, Lang Xiao, Junqi Che

**Affiliations:** 1College of Materials Science and Engineering, Chongqing University, Chongqing 400030, China; luo51520339@163.com (Z.L.); wangjing30353834@163.com (J.W.); 20173003@cqu.edu.cn (T.H.); 17623531908@163.com (L.X.); cjq342561227@163.com (J.C.); 2National Engineering Research Center for Magnesium Alloys, Chongqing University, Chongqing 400030, China; 3Chongqing Automotive Collaborative Innovation Center, Chongqing University, Chongqing 400030, China; 13883258439@163.com

**Keywords:** vacuum, porosity, high pressure die casting, tensile properties, fracture mechanism

## Abstract

The absolute pressure strongly affects the porosity and mechanical properties of castings produced by vacuum high-pressure die casting (V-HPDC) technology. The pore size, quantity and distribution of AlSi_9_Cu_3_ samples under three absolute pressures were evaluated by X-ray tomography and optical and electron microscopy. The paper presents an elaboration the stress concentration mechanism of pores affecting the tensile properties. According to a mathematical analysis of a sample under uniaxial stress, the greater the radius of the pore, the higher the stress value is at the pore perimeter. When the absolute pressure drops from 1013 mbar to 100 mbar, the porosity decreases from 6.8% to 2.8%, and the pore number and mean size decreases. In tensile tests, the pore sizes of the fracture surface decrease with decreasing absolute pressure, and the pore distribution becomes uniform. The tensile properties and extensibility of the sample are improved, and the microscopic fracture surface of the sample changes from cleavage fracture to quasi-cleavage fracture. The number, size and distribution of pores in die casting collectively affect the properties of the sample. Large-size or complex pores or pores with concentrated distributions produce large stress concentrations, decreasing the strength of the metal.

## 1. Introduction

The technology of high-pressure die casting (HPDC) is a special kind of casting method in modern metal processing technology involving little or no cutting. Due to its high productivity and its high shape and dimensional accuracy, HPDC has been widely used in the automobile, naval and aerospace fields, among other fields. The aluminum-silicon alloy, due to its light weight, good comprehensive mechanical properties, high wear resistance and low thermal expansion coefficient, is considered the best material for engine cylinder blocks [1,2,3]. HPDC is characterized by high pressure and high speed, so the metal liquid inevitably undergoes gas entrainment when filling the die cavity. The resulting pore defects have a significant effect on the mechanical properties of alloy materials. The porosity not only reduces the effective stress area of the casting but also induces crack formation at local stress concentrations. These effects severely limit the elongation of the alloy [4,5,6,7,8,9]. Generally, the pores are formed by residual gas in the castings and the gas involved in the filling process of a metallic liquid [10,11]. Vacuum die casting has attracted extensive attention as an advanced die casting technology that uses vacuum technology to remove air in the mold cavity for the purpose of reducing the casting porosity and enhancing the mechanical properties [12,13,14,15,16]. Compared with ordinary die casting, vacuum die casting has the following advantages: (1) reducing porosity in castings; (2) improving the finish of the casting surface; (3) improving the dimensional accuracy of castings and (4) obtaining castings with uniform microstructure [17]. Therefore, by reducing the gas entrapment of metal liquid in the filling process, we can obtain better-quality die casting products by vacuum die casting. Worldwide research on this method has focused on the effect of vacuum application on aluminum (magnesium) alloy die castings as characterized by the porosity and by the microstructural and mechanical properties.

Cao et al. studied the porosity and mechanical properties of AlSi_9_Cu_3_ die castings produced under different degrees of vacuum [18]. In that study, a V-HPDC process was used to produce AlSi_9_Cu_3_ alloy castings under three different absolute pressures of 500 mbar, 200 mbar and 100 mbar. The studies showed that as the absolute pressure decreased from 500 to 100 mbar, the average porosity decreased, and the elongation and tensile strength increased substantially. M. Koru et al. studied the effects of heat, dynamic parameters and vacuum application on porosity in HPDC of the A383 aluminum alloy [19]. Flow-3D analysis showed that the temperature, pressure and optimal dynamic parameters (velocity and vacuum) were the factors most affecting the mechanical properties and porosity of the samples. Compared with the results of the non-vacuum condition, the tensile strength and yield strength of samples cast under vacuum were improved. With increasing die temperature and the application of vacuum, the porosity decreased considerably. Increasing the injection rate led to increasing porosity and weakening of the mechanical properties, which could be mitigated by increasing the injection pressure. A. Zyska et al. studied the effect of applying vacuum to the cavity of an HPDC machine on the porosities of AlSi_9_Cu_3_ alloy castings [20]. With the other process parameters kept constant, three pressure values of 300 mbar, 500 mbar and 700 mbar were selected for testing, and the porosities of the castings were evaluated according to density measurements, microstructure observation and the hydrostatic weighing method. The results showed that the porosity decreased linearly with increasing degree of vacuum applied to the mold cavity. X.P. Niu et al. investigated the effects of vacuum assistance on the porosity distribution and mechanical properties of castings [21]. Three kinds of aluminum alloy samples (Al-5% Si, Al-8% Si and Al-18% Si) were selected for the experiment to be produced on a die casting machine equipped with a Fondarex vacuum auxiliary system. The research showed that vacuum-assisted casting strongly reduced the porosity of die castings and increased their density and mechanical properties of tensile strength and elongation. The numerical simulation of vacuum die casting can be used to improve the prediction accuracy of the defect position and elongation of die castings; in these studies, data have been obtained by simulation tests and verified by experiments [22,23,24,25,26].

The above research indicates that the application of a vacuum system in the production process of HPDC has a great impact on the quality of die castings, and substantial research on the pores and mechanical properties of aluminum (magnesium) alloy die castings in vacuum applications has been conducted by scholars worldwide. Specifically, vacuum application can reduce the porosity of die castings and improve their mechanical properties. However, the specific relationship between the absolute pressure, porosity and mechanical properties in the die casting process and the underlying mechanism of material fracture are not clear and need further study. 

In this paper, using experimental methods, the porosity and mechanical properties of AlSi_9_Cu_3_ die castings are evaluated in a comprehensive way under three different absolute pressures to reveal potential relationships between the absolute pressure, porosity and mechanical properties. In addition, the mechanism of influence of die casting pores on the mechanical properties of die casting samples is analyzed from the perspective of the material fracture mechanism. Through this study, we can better understand the effect of absolute pressure on the porosity and mechanical properties of die castings.

## 2. Materials and Methods

### Experiments

The die casting parts studied in this experiment were engine blocks, and the material was AlSi_9_Cu_3_ (Table 1). X-ray fluorescence spectrometer (XRF-1800 built by Shimadzu Corporation, Beijing, China) were used to determine the compositions of the alloy (AlSi_9_Cu_3_) in Chongqing University. The parts were produced by a horizontal cold chamber die casting machine (Buhler Evolution machine built by L.K. Technology Holdings Limited, Hong Kong, China) equipped with a vacuum-assistance system. The vacuum pump and the vacuum valve in this system work in conjunction to discharge the air contained in the injection cylinder, as well as the air remaining in the mold cavity of the castings. To ensure the feasibility of the experiment, the following process parameters were controlled: The die temperature and the casting temperature were set to 250 °C and 650 °C, respectively. The maximum wall thickness and the minimum wall thickness were 30 mm and 3.5 mm, respectively. The velocities of slow injection and fast injection were set to 0.2 m/s and 5 m/s, respectively. The absolute pressure was set to one of three values: 1013 mbar, 200 mbar or 100 mbar.

Samples (OM samples, SEM samples and tensile samples) were all taken from the produced engine cylinder blocks by wire-electrode cutting. The three-dimensional diagram of engine cylinder blocks is shown in Figure 1 (the cutting location of samples are shown by the red circle). The location of selected samples is the final solidification place during die casting simulation. The porosity and shrinkage cavity are easy to occur in this location, which is the defect feature focused on the research. Therefore, this location was selected to compare the microstructure and mechanical performance of samples under different absolute pressure levels. The microstructure, tensile test results and fracture morphology of the samples were evaluated. Optical microscopy (OM, model AXIOVERT 40 MAT, built by Carl Zeiss AG, Jena, Germany) was used to obtain a metallographic diagram of the samples by photography for pore analysis. X-ray industrial inspection equipment (UNC130 built by UNICOMP Holdings Limited, Chongqing, China) was used to identify the locations of pore defects. Scanning electron microscopy (SEM, model TESCAN VEGA II built by TESCAN, Brno, Czech Republic) was used to observe the microstructure of the samples and the pore morphology.

The dimensions of samples for the density measurements are 10 mm × 7.5 mm × 5 mm. The porosity rate of the casting was evaluated according to the standard BN-75/4051-10 [27]. The density was measured using the Archimedes method; the sample surface was kept clean and dry before being measured, the sample was weighed first in water and then in air, and the alloy density was calculated by Equation (1).
(1)$$\rho_{p}=\frac{m_{1}}{m_{1}-m_{2}}\cdot \rho_{w}$$
where: $\rho_{p}$ is the density of the specimen, $m_{1}$ is the mass of the specimen in the air, $m_{2}$ is the mass of the specimen in the water and $\rho_{w}\;$is the density of the water.

Next, the porosity of the specimen is calculated by Equation (2):(2)$$P=\left(1-\frac{\rho_{p}}{\rho_{wz}}\right)\cdot 100\%$$
where: $\rho_{wz}\;$is the standard density of EN AB 46000 according to EN1706, which is 2.76 $g/{\mathrm{cm}}^{3}$.

A tensile testing machine (10T electronic universal material testing machine built by INSTRON CORPORATION, Boston, America) was used to measure the tensile strength of the die casting samples at room temperature. The tensile rate is 1 mm/min. The dimensions of the tensile sample with rectangular cross-section are shown in Figure 2.

## 3. Results and Discussion

### 3.1. Calculation of Density and Porosity

The dimensions of samples for OM were 10 mm × 7.5 mm × 5 mm. The samples were used for OM after coarse grinding, accurate grinding and buffing (no metallographic corrosion). The microstructure and pore defects were visualized by assembling a composite image of 16 OM images captured at a magnification of 50×, as shown in Figure 3.

The number and size of the pores of each sample were counted with a NanoMeasurer 1.2 software quantitative analysis system for metallographic characterization (Figure 3). The pore statistics of the sample are shown in Figure 4 and Table 2. The dimensions of the sample pores under the three different vacuum levels were distributed mainly within 4–12 μm. With decreasing absolute pressure, the number of plate surface pores gradually decreased; the largest and smallest diameter of the pores and the average diameter were also reduced. The number of pores was maximized at a vacuum level of 1013 mbar; there was no obvious trend of change at 200 mbar or 100 mbar. 

The porosity and density of the castings under the three different vacuum levels of 1013, 200 and 100 mbar are shown in Table 2. The porosity and density decreased and increased, respectively, with increasing degree of vacuum. The sample porosity and density of the sample with a vacuum level of 1013 mbar increased by 143% and decreased by 5.8%, respectively, compared with the results for 100 mbar.

Cao et al. discussed the effect of the absolute pressure in the cavity on the pore defects of die castings [28]. In the vacuum die-casting process, the formation of eddy currents during the filling of molten metal is the main cause of void defects. The reduction in the absolute pressure suppresses the generation of metal–liquid separation flow, making the vortex formation more difficult. The decrease in the number of vortices reduces the amount of gas involved before the metal solidifies, so the number of pore defects in the die casting is also reduced.

### 3.2. Characterization of Pore Defects

The pore defects were observed by OM and X-ray imaging (Figure 5). The pore defect distribution patterns at a fixed horizontal height were observed with the optical micrographs. The pore defect distribution patterns at different depths in the same direction were observed by the X-ray tomography. The X-ray images broke through the limitation of two-dimensional plane and avoided the contingency of selection area observed by OM. The dimensions of the sample for X-ray tomography are 10 mm × 7.5 mm × 5 mm and the scanning surface was the sample plane of 10 mm × 7.5 mm. As shown in Figure 5, the number and size of pore defects decreased significantly with decreasing absolute pressure (Figure 5b,d,f shows the distribution of pore defects from 1013 mbar to 100 mbar. The white light spots indicate the pore locations. The brighter the white light spot, the closer the pores is to the scanning plane).

### 3.3. Measurement of Tensile Strength and Elongation

The measurement data for the tensile tests are shown in Figure 6 and Table 3. As the absolute pressure decreased, the tensile strength and elongation increased. The brittle fracture occurred on metals and alloys of high strength or low ductility and toughness. On the other hand, a brittle fracture occurred in the presence of pore defects or initial cracks, even if the metal had good ductility. Defects and cracks produced stress concentrations that were several times higher than the average tensile stress. If the overconcentrated tensile stress exceeds the critical tensile stress value of the material, crack or defect propagation will occur, leading to brittle fracture. As can be seen from Figure 6 and Table 3, the elongation of the sample was low, so it could be inferred that the material belongs to the brittle fracture.

The fracture morphology and microstructure of tensile samples after fracture are shown in Figure 7. Figure 7a,c,e shows macro-morphological features of the casting fracture surfaces, none of which are significantly deformed. The surfaces are flat, which is typical for brittle fracture. With decreasing absolute pressure, the surface appearance of the sample fracture surface becomes even flatter. In addition, the fracture surfaces exhibit prominent casting defects, such as shrinkage pores, gas entrainment pores, and cracks. Figure 7a shows large pore defects on the surface of the fracture surface for the sample cast at 1013 mbar; the distribution is more concentrated (shown by the red circle) than at other pressures. In contrast, the fracture surfaces for 200 mbar and 100 mbar have many smaller and more uniformly distributed pores. During the tensile test, pores with a larger size or more numbers result in a higher overall stress that promotes a fracture.

Figure 7b,d,f shows the microscopic topographies of the casting fracture surfaces. As observed in Figure 7b, the microscopic appearance of the fracture is a typical representation of cleavage fracture, river patterns and small, flat planes. Microscopic cracks are also found on the fracture surface. Initial crack formation is generally related to plastic deformation; when a dislocation slip surface moves in a grain, dislocations are formed at the intersection of the slip planes, resulting in stress concentrations [29]. If this stress cannot be relaxed by other ways, an initial crack will form perpendicularly on the crystal surface, and the existence of the crack will greatly reduce the tensile strength of the casting. The sample prepared at 1013 mbar has the lowest tensile strength. With decreasing absolute pressure, additional slip bands can be observed in the microscale fracture appearance as shown in Figure 7d, appearing between the tongue patterns. These slip bands are tightly distributed in fine strips and are accompanied by additional regions of slip. Slip bands, which are common morphological features of quasi-cleavage fractures, are often considered the result of interaction between cleavage cracks and grains during plastic deformation. When the vacuum level increased to 100 mbar, slip bands appeared in the cross section, as shown in Figure 7f, and dimples appeared, which is typical of a quasi-cleavage fracture surface. The crack source for a quasi-cleavage fracture is usually inside the small quasi-cleavage section; compared with the path of the solution crack, the expansion of the quasi-cleavage crack is not continuous but instead corresponds to the local formation of cracks and local expansion, thus forming many small quasi-cleavage planes. Figure 7 shows that with increasing degree of vacuum, the sizes of the pores on the sample fracture surface are reduced, the pores are more uniformly distributed, and the microfracture morphology of the sample changes from that of the cleavage fracture to the quasi-cleavage fracture.

It can be observed from Figure 7 that there are many pores in the fracture morphologies. According to the calculation equation of tensile strength, the tensile strength is the ratio of the tensile force to the cross-sectional area of the sample. Therefore, the reduction of effective cross-sectional area caused by porosity defects is also one of the reasons for the reduction of tensile strength [30,31]. Under 1013 mbar，the pores on the fracture surface of the sample with a larger size, a higher number and uneven distribution, greatly reduced the effective cross-sectional area of the tensile sample, and thus led to the reduction of tensile strength. Under 200 mbar and 100 mbar, although there were pores on the fracture surface of the sample, they were smaller in size, less in number and more even in distribution. In this case, the effect of pores on tensile strength was less, so the tensile strength obtained by tensile test was higher than that of die casting under 1013 mbar.

### 3.4. The Stress Concentration Mechanism of Tensile Fracture

The mechanical effects related to the fracture in the tensile test also merit study. The presence of pores inside the metal microstructure cause stress concentrations in the areas surrounding the pores. Idealizing the pore as a circle, G. N. Savin [32] discussed the mathematical problem of stress concentrations near a pore. The plane problem of elasticity theory involves integrating the equilibrium and compatibility equations under the corresponding boundary conditions. The equilibrium and compatibility equations are shown in Equations (3) and (4), respectively.
(3)$$\frac{\partial \sigma_{x}}{\partial x}+\frac{\partial \tau_{zy}}{\partial y}=0;\frac{\partial \tau_{zy}}{\partial x}+\frac{\partial \sigma_{y}}{\partial y}=0$$
(4)$$\left(\frac{\partial^{2}}{\partial x^{2}}+\frac{\partial^{2}}{\partial y^{2}}\right)\left(\sigma_{x}+\sigma_{y}\right)=0$$

The stress function is introduced, and the former equation can be represented in the form of a two-component function, as shown in Equation (5).
(5)$$U\left(x,y\right)=Re\left[\bar{z}\varphi_{1}\left(z\right)+\chi_{1}\left(z\right)\right]$$

Re represents the real part of the expression in square brackets; $\varphi_{1}\left(z\right)$ and $\chi_{1}\left(z\right)$ are the parsing functions z=x+iy and $\bar{z}=x-iy$ of the complex variable, respectively.

According to the Kolosoff–Muskhelishvili equation, $\sigma_{x}$, $\sigma_{y}$ and $\tau_{xy}$ can be expressed in the form of Equation (6).
(6)$$\sigma_{x}+\sigma_{y}=2\left[{\varphi^{\prime }}_{1}\left(z\right)+\bar{{\varphi^{\prime }}_{1}\left(z\right)}\right];\sigma_{x}-\sigma_{y}+2i\tau_{xy}=2\left[\bar{z}\varphi^{\prime \prime }{}_{1}\left(z\right)+\bar{{\;\psi \;\prime \prime }_{1}\left(z\right)}\right]$$

The area S is mapped within the unit circle (or outside the circle) by the mapping function z=ω(ξ), with σ representing the value of variable ζ on the unit circle. After replacing the variable z with ω(σ), Equation (6) can be expressed as Equation (7) (represented in polar coordinates): (7)$$\sigma_{\theta }+\sigma_{\rho }=2\left[\Phi \left(\xi \right)+\bar{\Phi \left(\xi \right)}\right]\;\sigma_{\theta }-\sigma_{\rho }+2i\tau_{\rho \theta }=\frac{2\xi^{2}}{\rho^{2}\omega^{\prime }\left(\xi \right)}\left[\bar{\omega \left(\xi \right)}\emptyset \prime \left(\xi \right)+\omega^{\prime }\left(\xi \right)\Psi \left(\xi \right)\right]\;\Phi \left(\xi \right)=\frac{\varphi '\left(\xi \right)}{\omega '\left(\xi \right)};\Psi \left(z\right)=\frac{\psi \prime \left(\xi \right)}{\omega \prime \left(\xi \right)}$$

The stress function is shown in Equation (8):(8)$$\varphi \left(\xi \right)=\frac{X+iY}{2\pi \left(1+x\right)}\ln\xi +c\cdot \frac{B+iC}{\xi }+\varphi_{0}\left(\xi \right);\psi \left(z\right)=\frac{x\left(X-iY\right)}{2\pi \left(1+x\right)}\ln\xi +c\cdot \frac{B_{1}+iC_{1}}{\xi }+\psi_{0}\left(\xi \right)$$

Here, $\varphi_{0}\left(\xi \right)=\overset{\infty }{\underset{0}{\sum }}a_{n}\xi^{n}$ and $\psi_{0}\left(\xi \right)=\overset{\infty }{\underset{0}{\sum }}b_{n}\xi^{n}$ are two regular functions of the complex variable ξ within the unit circle.

If a plate with a circular pore with a radius of R is stretched by an external force P along the O_z_ axis, the stress function is as shown in Equation (9).
(9)$$\varphi \left(\xi \right)=\frac{pR}{4}\left(\frac{1}{\xi }+2\xi \right);\psi \left(\xi \right)=-\frac{pR}{2}\left(\frac{1}{\xi }+\xi -\xi^{2}\right)$$

Here, the larger the radius R of the circular pore, the greater is the corresponding stress value. The initial stress cracks occur in the region where the stress is concentrated during the stretching process (that is, where large pores appear), which leads to a fracture.

Substituting the values of related functions in Equation (9) into Equation (7), the stress functions $\sigma_{\rho },\;\sigma_{\theta }\;\mathrm{and}\;\tau_{\rho \theta }$ are obtained as shown in Equation (10).
(10)$$\sigma_{\rho }=\frac{p}{2}\left[\left(1-\rho^{2}\right)+\left(1-4\rho^{2}+3\rho^{4}\right)\cos2\theta \right]\;\sigma_{\theta }=\frac{p}{2}\left[\left(1+\rho^{2}\right)-\left(1+3\rho^{4}\right)\cos2\theta \right]\;\tau_{\rho \theta }=\frac{p}{2}\left[\left(1+2\rho^{2}-3\rho^{4}\right)\sin2\theta \right]$$

When ρ=1.0, 0.9, 0.8, 0.7, 0.5 and 0.3, the values of $\sigma_{max},\;\sigma_{min}\;\mathrm{and}\;\tau_{max}$ can be calculated according to Equation (11).
(11)$$\sigma_{\begin{array}{c} max\\ min \end{array}}=\frac{\sigma_{\rho }+\sigma_{\theta }}{2}\pm \sqrt{{\left(\frac{\sigma_{\rho }+\sigma_{\theta }}{2}\right)}^{2}+\tau_{\rho \theta }^{2}};\tau_{max}=\pm \sqrt{{\left(\frac{\sigma_{\rho }+\sigma_{\theta }}{2}\right)}^{2}+\tau_{\rho \theta }^{2}}$$

The principal stress trajectory is plotted with Equation (10).
(12)$$\mathrm{tg}2\alpha =\frac{2\tau_{\rho \theta }}{\sigma_{\rho }+\sigma_{\theta }}$$

According to Equations (10)–(12), the isostress parameters $\tau_{max}$ and $\sigma_{max}$ and $\sigma_{min}$ are drawn in Figure 8; the trajectory of principal stress is drawn in Figure 9.

As can be seen from Figure 8 and Figure 9, the sample had the highest stress concentration in the horizontal direction of the pore boundary under uniaxial tensile stress, and the larger the pores, the greater the stress value. As a result, cracks initiated at the boundary in the horizontal direction of large pores and led to breakage.

It has been noted in the literature [33,34,35,36] that crack initiation during the sample fracture is related to the pores inside the sample and that the crack source can be found near the boundary of the sample by analyzing the sample fracture surface. The larger the diameter of the pore, the more casting fatigue cracks initiate and propagate. The size and distribution of the pores collectively affect the performance of the sample. Pores with a larger size and higher pore densities produce larger stress concentrations, leading to sample fracture at the crack location. The macroscopic and microscopic morphologies of the sample fractures are in accordance with the variation trend of the mechanical properties of the castings. Therefore, it can be inferred that decreasing the absolute pressure can effectively reduce the porosity and the pore size of a casting, which can reduce stress concentration near the pores and improve the fatigue life of the casting.

## 4. Conclusions

To study the influence law of die casting pores on the mechanical properties of die castings, OM and X-ray were used to capture a metallographic map and analyze the number, size and distribution of die casting pores at different absolute pressures. SEM was used to obtain micrographs of tensile fracture surfaces. Tensile experiments were conducted on samples. Then, the stress concentration mechanism of the tensile fracture was evaluated.

(1) From the test results, the density of the die castings increased with decreasing absolute pressure, and the porosity of the die castings and the average pore size decreased.

(2) From the tensile test results, the tensile strength and elongation of the die casting samples increased with decreasing absolute pressure. With decreasing absolute pressure, the surface appearance of the sample fracture surface became flatter, the sizes of the surface pores were reduced, the pore distribution became more uniform and the fracture type changed from a cleavage fracture to quasi-cleavage fracture.

(3) From the results of mathematical model analysis, pore defects had an important effect on the tensile performance of die castings. The larger the pore size, the greater the corresponding stress value in the horizontal direction of the pore boundary. The stress concentration mechanism near pores caused initial cracks and crack propagations around the pores during the tensile process, which led to the decrease of tensile strength.

## Figures and Tables

**Figure 1 materials-13-03019-f001:**
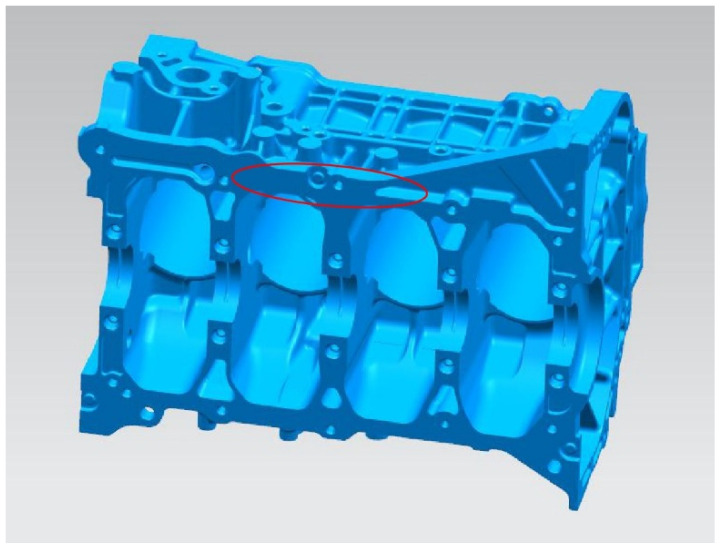
The three-dimensional diagram of engine cylinder blocks.

**Figure 2 materials-13-03019-f002:**
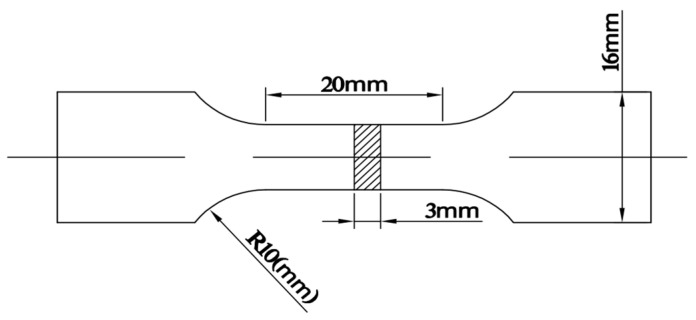
The dimensions of the tensile sample.

**Figure 3 materials-13-03019-f003:**
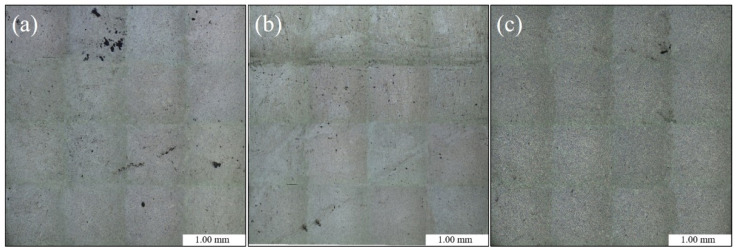
Full microscope images of samples under different degrees of vacuum: (**a**) 1013 mbar, (**b**) 200 mbar and (**c**) 100 mbar.

**Figure 4 materials-13-03019-f004:**
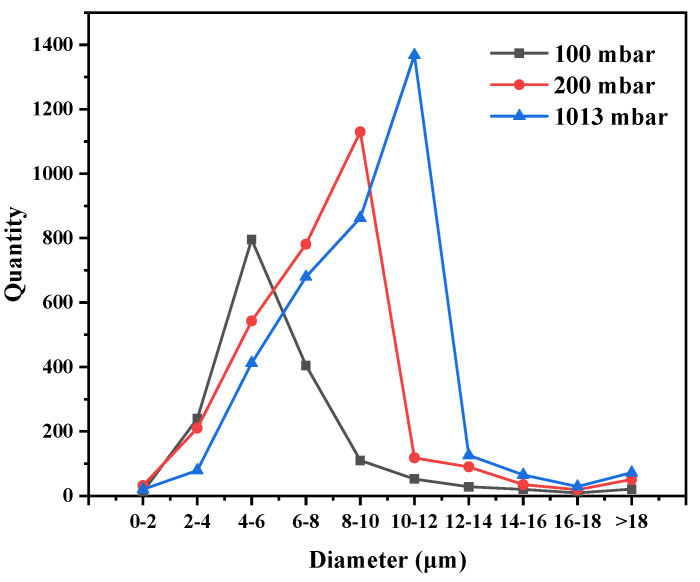
The calculation results of the pore numbers and dimensions.

**Figure 5 materials-13-03019-f005:**
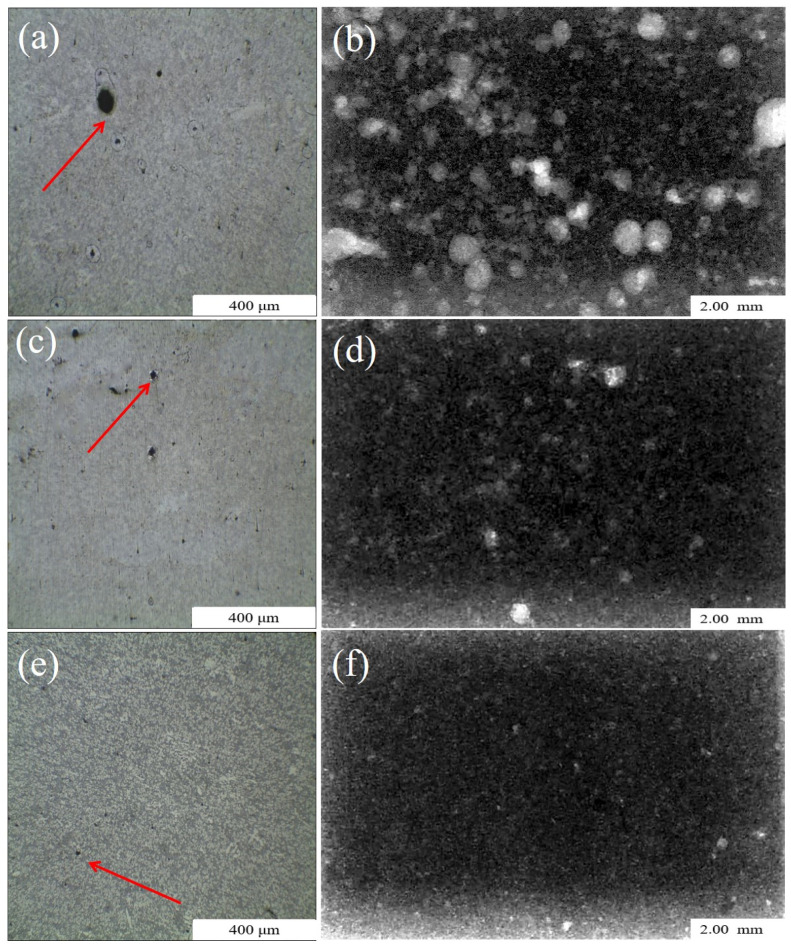
Comparison of porosity microstructure in samples: (**a**) OM image at 1013 mbar; (**b**) X-ray image at 1013 mbar; (**c**) OM image at 200 mbar; (**d**) X-ray image at 200 mbar; (**e**) OM image at 100 mbar; (**f**) X-ray imaging at 100 mbar.

**Figure 6 materials-13-03019-f006:**
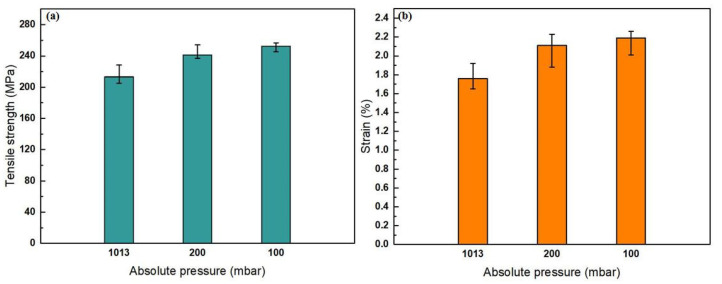
Tensile strength and elongation at different absolute pressures: (**a**) tensile strength and (**b**) elongation.

**Figure 7 materials-13-03019-f007:**
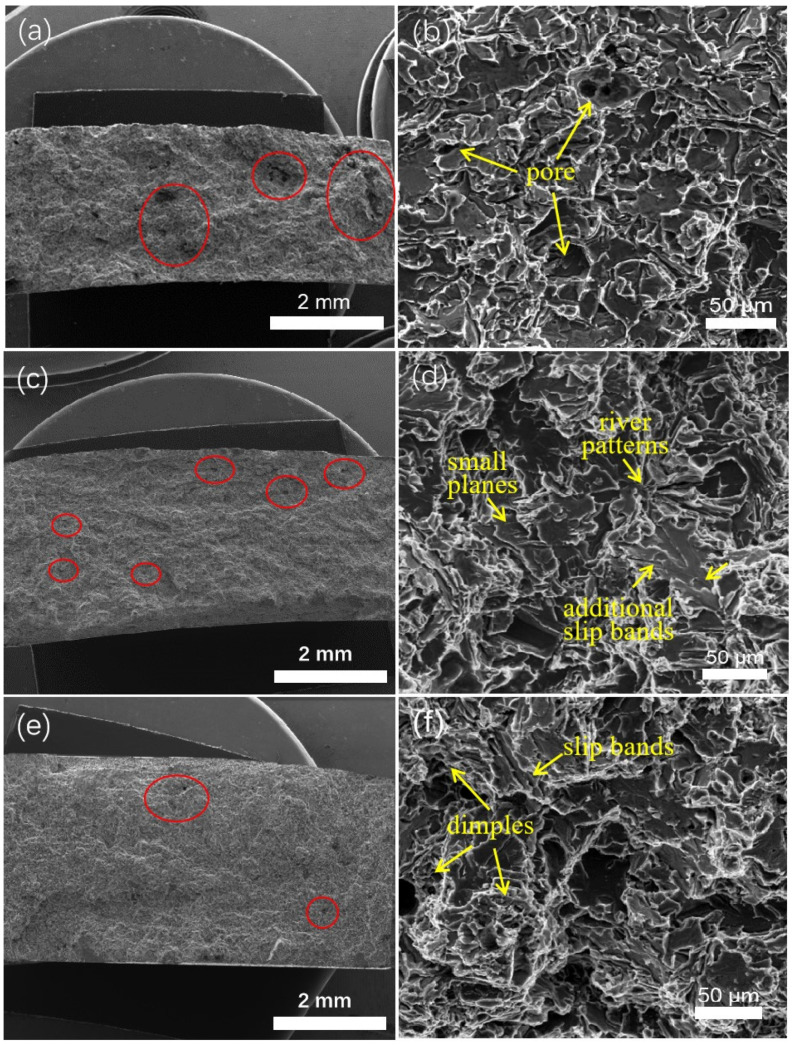
SEM images of tensile fracture morphologies of samples cast under different absolute pressures: (**a**), (**b**) 1013 mbar; (**c**), (**d**) 200 mbar and (**e**), (**f**) 100 mbar.

**Figure 8 materials-13-03019-f008:**
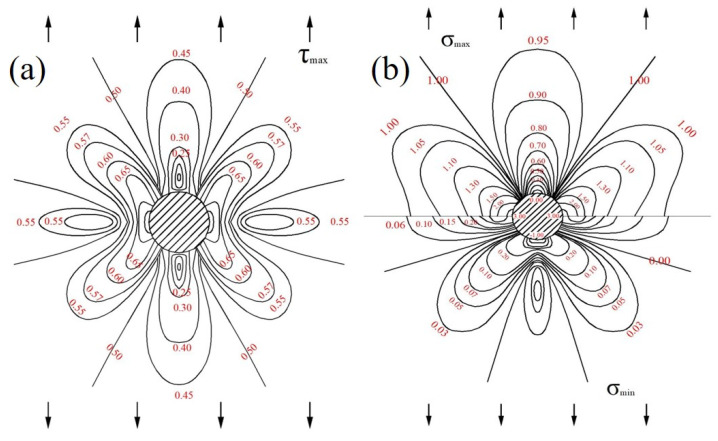
Isostress diagram of circular pores under tension: (**a**): $\tau_{max}$ and (**b**): $\sigma_{max}$ and (**c**): $\sigma_{min}$.

**Figure 9 materials-13-03019-f009:**
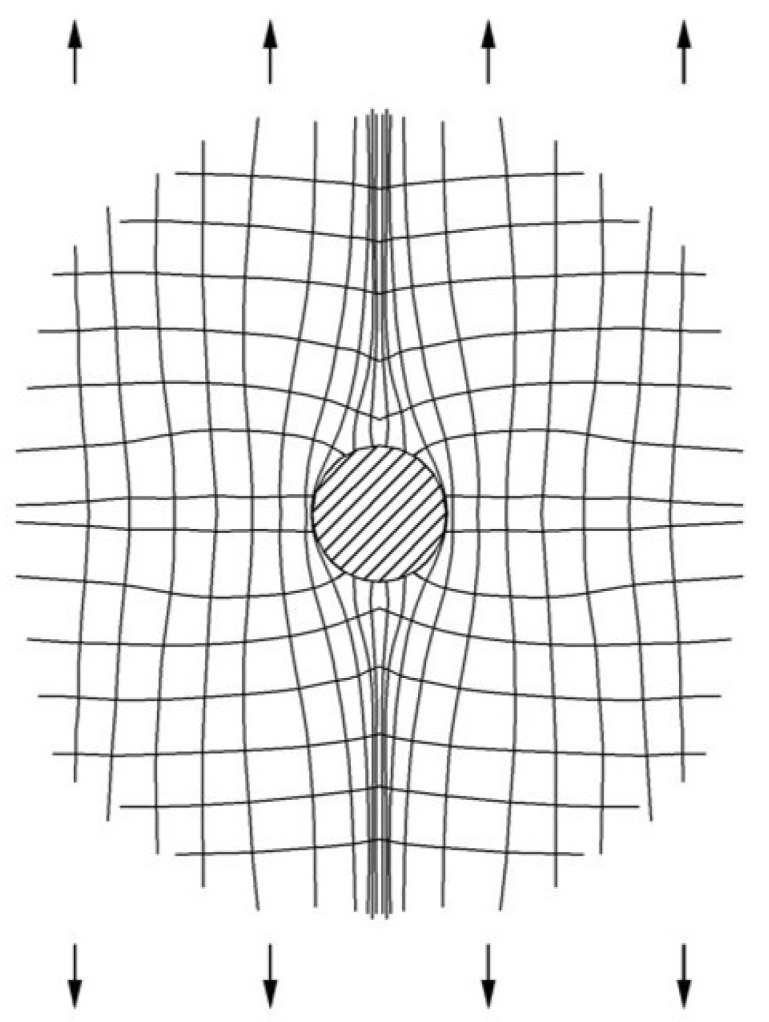
Principal stress trajectory of a circular pore under tension.

**Table 1 materials-13-03019-t001:** The X-ray fluorescence spectrometer (XRF) results of chemical composition limits of AlSi_9_Cu_3_ (wt %).

Element	Si	Cu	Mn	Zn	Fe	Al
wt %	10.03	2.92	0.32	1.00	0.71	REM

**Table 2 materials-13-03019-t002:** The results of the porosity calculations for the die castings.

Absolute Pressure (mbar)	Number	Average Size (μm)	Porosity (P) (%)	Maximum Diameter (μm)	Minimum Diameter (μm)	Density (g/cm^3^)
1013	3714	11.65	6.8	142.3	2.24	2.526
200	3009	8.12	4.4	127.59	1.98	2.605
100	1694	5.61	2.8	111.9	1.44	2.682

**Table 3 materials-13-03019-t003:** Tensile test results of samples cast at different absolute pressures.

Absolute Pressure	Stress (MPa)	Elongation (%)
1013 mbar	213.14	1.76
200 mbar	241.09	2.11
100 mbar	252.28	2.19

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
