# Peer review of "The Stress Concentration Mechanism of Pores Affecting the Tensile Properties in Vacuum Die Casting Metals"

_materials, 2020, doi:10.3390/ma13133019_

Round 1
Reviewer 1 Report
The paper was interesting for a number of reasons. Overall, the authors achieved their stated intent and the paper is worthy of publication. There are some careless mistakes throughout that need to be caught (partly significant figures in tables, see below for other concerns).
The analytical/theoretical section needs to be cleaned up. There are a lot of equations, some of which could be eliminated, but at the very least the numbering must be checked as there are duplicates and numbering errors throughout.
Also, some word usage needs to be checked (like "germinate" (biology term) used for initiation of a crack).
Finally, a personal preference item. In the conclusions, first paragraph, the authors (for some reason) started using "we" instead of just stating what was done. "... we used OM and X-ray...", etc. If you insist on opening the conclusions this way, please change to state what was done, like " ... OM and X-ray were used to capture ..." or "We used SEM to obtain ..." to "SEM was used to obtain ..."
From a technical perspective related to sharing information more widely, I would like to offer the following for consideration. While the standard alloys under consideration are well known it might still be useful to fully describe the chemistry of the ones tested in your study as major, minor and trace element control is critical to not only the manufacturing process but also the physical and mechanical properties.
Also, the manufacturing method used in the experimental design (fixed and variable) were very well spelled out. The sectioning and preparation of tensile specimens (and other microstructural sectioning for density and porosity) could have been described more completely. It is not clear if the microstructural information came from the same chunk of material used for tensile specimens. If so, it should be stated that this was the case. In addition, there was a reason for taking specimens from this location, and again, that reason should be explained. Also, (understanding funding, effort and the like) is the area sampled in the engine block typical of all areas (and if so how determined, even if it is other work, published or not)? If so, then this should be shared as well.
In addition, the tensile specimens description could be improved (I assume a round bar was used but it was not explicitly stated). I a little concerned that the tensile results appear to be from one test. One test for a casting! My work with castings indicate a level of scatter in the data. So ... if this is not the case, then this too should be explained and any statistics should be included (or the use of just one specimen qualified and put into context). If just one tensile test for each manufacturing route was used, then the authors need to support that with something else (like hardness for other regions or visual inspection or ???).
The work done on pore size was pretty impressive but other statistics could improve the validity of the results and strengthen the conclusions (like number of fields captured and analyzed, number pores in each bin for each pressure level, average and standard deviation for each bin for each pressure level, etc.).
Without belaboring the point, I am suggesting that while these results look interesting and are what I would have expected, details are very important when considering the validity and quality of the work. Part of the work is very high quality and other parts don't seem to be. I am suggesting that attention to detail is very important in a paper like this one (all the details where it doesn't intrude on intellectual property concerns) and more can be done. However, it is not my intent to suggest the authors do more work but if such work has already been done, then they might consider including it in this paper. At the very least, explanation regrading these concerns need to be included within the paper to put into context the validity of the theoretical work as it is used to support the conclusions. It would make it much better than it is.
My best regards.
Reviewer 2 Report
The paper has a good narrative but is written in first person in places and there are some grammatical and spelling errors. The overall conclusion appears to be that reducing the vacuum casting pressure decreases porosity and thus increases tensile strength, however there is little discussion of the results. There are also some inconsistencies in the paper and some points need clarifying. These are outlined below.
In the experimental procedure section, why was the region chosen for the analysis? Some more information about this region, such as wall thickness is required as porosity will be dependent on geometry. Was the porosity evenly distributed around the cast part? The authors should indicate what size sample was used for the density measurements.
The authors note the use of x-ray tomography. The x-ray images, shown in figure 5, represent a single image of each sample and not a series of slices or 3D reconstruction. Furthermore, according to the scale bar, the pores in these images are up to 1mm in diameter, much larger than identified elsewhere in the paper.
The tensile curves shown in figure 6 need more description as they are dissimilar in shape to other similar alloys. Are these plots true stress/strain or engineering stress/strain? The initial gradients for 100mbar and 200mbar are different and at these low values of strain, these differences are not related to fracture.
The fracture surfaces shown in figure 7 are rectangular but the tensile samples, according to figure 2, have a diameter of 6mm. If these have been sectioned, then this should be clarified. The max dimension of these fracture surfaces seems to be in excess of 6mm. The authors have identified different features on the fracture surfaces for pressures of 200mbar and 100mbar however, figures 7c and 7e are identical. This raises questions about the integrity of the results.
The calculation of stress field around the pores, although interesting, adds little to the paper. Finite element modelling of the stress field around the pores could be used to compare to the analytical results and would enhance the paper. Based on the non-linear shape of the stress/strain curve, the stress would redistribute around the pores during deformation.
The last paragraph of the discussion mentions the effect of porosity on fatigue crack propagation and growth, however the presented work focuses on tensile properties.
The first and second conclusions are supported by the research, but the third conclusion is not.
Based on the above comments, I would recommend that the paper is thoroughly rewritten before re-submission with much clearer discussion and conclusions.
Reviewer 3 Report
I recommend the article entitled: „The Stress Concentration Mechanism of Pores Affecting the Tensile Properties in Vacuum Die Casting Metals” for publication in the Materials journal after making the recommended corrections. The proposed amendments are generally of editorial nature. The data presented in Figure 7 are the most doubtful. Figures 7c and 7e represent the same microstructure, but the content of the article shows that these are microstructures obtained at different absolute pressures. In addition, it is worth highlighting in Fig. 7 b, d and f the fracture features described in the content of the article (e.g. river patterns, small, flat planes, or slip bands). The data provided in Fig. 7 will be much easier for the potential reader to see. Detailed comments are included in the text of the article in the attached file.

Round 2
Reviewer 1 Report
Very much improved.
Author Response
Thank you for your review.
Reviewer 2 Report
The amendments made in the revised manuscript improve the presentation of the research to readers I recomend that this revison be accepted. However, there are a few points that need further clarifying.
Firstly, I see that the specimen geometry has changed and that the unusual shaped tensile curves have been removed. The new presentation of the tensile results include standard deviations. How were these calculated? If more than one test was completed, this should be stated. Also, the methods section does not include details about how strain was measured. Was a strain gauge used or was strain calucated from crosshead displacement? If the latter, this may help to explain the shapes of the curves observed in the previous revision. I think that the brittle nature of fracture should be highlighted more, since this is important for the next section on stress concentration.
Also, one of the main conclusions relates to pore radius and its effect on tensile strength. It sould be clarified that this relates to pore size. Previous work on notches has shown that notches with smaller radii result in higher stress concentrations for equivalent notch depths.
